# Towards High Efficiency and Rapid Production of Room-Temperature Liquid Metal Wires Compatible with Electronic Prototyping Connectors

**DOI:** 10.3390/mi14122227

**Published:** 2023-12-12

**Authors:** Luka Morita, Shima Jalali, Abolfazl Vaheb, Rawan Elsersawy, Kunj Golwala, Asad Asad, Patricia I. Dolez, James D. Hogan, Mohammad Abu Hasan Khondoker, Dan Sameoto

**Affiliations:** 1Department of Mechanical Engineering, University of Alberta, Edmonton, AB T6G 1H9, Canada; lmorita@ualberta.ca (L.M.); shima1@ualberta.ca (S.J.); kunjneha@ualberta.ca (K.G.); aaasad@ualberta.ca (A.A.); jdhogan@ualberta.ca (J.D.H.); 2Industrial Systems Engineering, University of Regina, Regina, SK S4S 0A2, Canada; rre285@uregina.ca; 3Department of Human Ecology, University of Alberta, Edmonton, AB T6G 2N1, Canada; pdolez@ualberta.ca

**Keywords:** EGaIn, gallium, interconnection, SEBS, extrusion, thermal drawing, sensors

## Abstract

We present in this work new methodologies to produce, refine, and interconnect room-temperature liquid-metal-core thermoplastic elastomer wires that have extreme extendibility (>500%), low production time and cost at scale, and may be integrated into commonly used electrical prototyping connectors like a Japan Solderless Terminal (JST) or Dupont connectors. Rather than focus on the development of a specific device, the aim of this work is to demonstrate strategies and processes necessary to achieve scalable production of liquid-metal-enabled electronics and address several key challenges that have been present in liquid metal systems, including leak-free operation, minimal gallium corrosion of other electrode materials, low liquid metal consumption, and high production rates. The ultimate goal is to create liquid-metal-enabled rapid prototyping technologies, similar to what can be achieved with Arduino projects, where modification and switching of components can be performed in seconds, which enables faster iterations of designs. Our process is focused primarily on fibre-based liquid metal wires contained within thermoplastic elastomers. These fibre form factors can easily be integrated with wearable sensors and actuators as they can be sewn or woven into fabrics, or cast within soft robotic components.

## 1. Introduction

The rapid advancement of flexible and stretchable electronics has sparked significant interest in the use of liquid metal alloys as a means to achieve metallic conductivity, with extreme mechanical deformations. Eutectic gallium–indium (EGaIn), with its low melting point, non-toxicity, and high electrical conductivity, has emerged as a promising liquid metal alloy for interconnecting electronic components in various stretchable and wearable applications due to its low toxicity compared to mercury, negligible vapour pressure, and unique formation of a nanoscale thickness oxide layer that works to stabilize the alloy into various microstructured features using direct write, casting, dipping, or injection techniques [1,2,3,4,5]. The ability of EGaIn to flow and conform to complex shapes enables seamless integration with flexible substrates, facilitating the development of conformal and stretchable electronic systems. The vast majority of reported work on the use of EGaIn usually centres on developing a new specific device [6,7,8,9], or a patterning process for the liquid metal on 2D substrates [3,7,10,11,12], similar to microfluidics, microelectronics, or microelectromechanical systems (MEMS). Less work has been done with specific attention paid to interconnection [13,14,15], packaging, leak-free operation [8,15], and scalable manufacturing production [2,3]. In particular, the integration of EGaIn interconnections with sensors and actuators holds great promise for the advancement of wearable devices [16], soft robotics [17], and biomedical applications [18]; however, in the vast majority of cases, these connections are specific to each device, and the liquid metal components cannot be repurposed or reused easily. Some recent work describes using alternative materials other than silicone rubbers for recoverable devices [19] as a means to reduce potential e-waste, but most reported designs in that work use a substantial amount of material and a multi-step recycling and separation process consisting of solvent dissolution of polymers, acid or base treatment, and either magnetic or electrochemical separation of the e-waste. While the goal of reducing e-waste is excellent, the complication of the breakdown of materials is still substantial, and repurposing materials would take several days. Being able to repurpose or reprogram liquid metal devices as easily as systems based on Arduino electronics could both aid in prototyping as well as avoid the need to complexly break down a device after use to get access to the base materials. To achieve this, better manufacturing and electromechanical interconnections are required; these were a key motivation for this paper.

One of the main challenges in liquid metal interconnections lies in achieving reliable and robust interfaces between EGaIn and other mechanical components, which are often orders of magnitude stiffer than the elastomer substrates that may contain liquid metals for stretchable electronics. Several major challenges in the field have been identified in recent review articles [10,20]. These challenges include the leaking of liquid metal in high-pressure applications, corrosion of metallic electrodes in contact with Gallium alloys, large-scale production with high consistency between devices, and using as little material as possible for functional devices to reduce costs (the commodity gallium, for example, is volatile in pricing and can cost over 500 USD/kg based on recent history at the time of writing) [21]. For academic work, the primary costs in manufacturing devices are based on researcher time, and occasionally the other polymers like polydimethylsiloxane (PDMS) that are used in the production of stretchable devices. Hand manufacturing/assembly, however, is not scalable and the soft lithography process and EGaIn fabrication like injection into microchannels become less and less practical as the number of devices needed for production increases, or the minimum features become so small the metal cannot fill without very high pressures. While the vast majority of elastomers reported to be integrated with EGaIn and other liquid metal alloys in the academic literature have been based on silicone rubbers, there are several reported works with thermoplastic elastomers as the encapsulation material [7,18,19,22,23,24]. These materials combine many of the advantages of silicone rubbers (high extensibility, resilience) with the processing advantages of thermoplastics, such as being easily moulded, embossed, extruded, and welded together with heat or solvents. As such, they offer a promising alternative for the scalable manufacturing of liquid-metal-enabled sensors and actuators. Especially in the field of wearable components, where certain advantages of silicone, such as extremely high temperature tolerances, are not expected to be needed, thermoplastic elastomers may hold a key advantage in the leap to commercial products.

As a step in the direction toward more “plug-and-play” systems that use the advantages of EGaIn and other liquid metal alloys, we present improvements on our earlier reported work on direct extrusion of liquid-metal-core elastomers [22]. We are working towards a continuous production process that can be post-processed to sever liquid metal within thermoplastic elastomer shells, seal those same shells to prevent leaking, and provides several options to connect liquid metal cores to solid metal electrodes with lower chances of corrosion or formation of intermetallic compounds. The utility of these liquid metal wires at a size scale that is much smaller than the systems we have presented earlier allows small systems to be designed with the same rapid prototyping philosophy that has been so useful in Arduino electronics designs, such that components may be reusable and recoverable as iterative design is pursued. Simple and effective interconnections to normal electronic components would not require substantial technical skills or expensive equipment investments. By focusing on fibre form factors, it is our hope that these systems eventually make it into smart clothing and fabrics [25,26,27,28], wearable computers/sensors [16], sensorized materials [29], or artificial muscles [30] for future wearable applications for soft robotic systems. This work demonstrates for the first time strategies for the continuous production of liquid metal wires and their subsequent reduction in size to create microscale features that are still compatible with standard electronics prototyping connectors. In particular, the ability to have larger diameter liquid metal core rubber wire of substantial length cut and sealed directly without loss of internal liquid metal is extremely important for eventual mass production as hand injection or manual operations can be minimized or eliminated with future development.

## 2. Materials and Methods

### 2.1. Materials

Styrene-ethylene-butylene-styrene (SEBS) G1657 and G1645 were obtained from Formerra Distribution Canada (Mississauga, ON, Canada) (formerly PolyOne) and used as received. Gallium and indium were obtained from rotometals.com (San Leandro, CA, USA) and combined at 75.5% wt% gallium to 24.5% wt indium by melting the gallium in a container surrounded by boiling water and adding indium while stirring until no solid metal remained. The oxide was avoided by extracting material from the sub-surface of the alloy with a syringe before use. Corrosion-resistant tungsten wire 0.025” in diameter (part # 3775K27) and polyolefin heat shrink tubes (part # 7496K81) were purchased from McMaster-Carr (Elmhurst, IL, USA).

A Wellzoom B extruder was purchased from Amazon.com (Bellevue, WA, USA). The nozzle was replaced with a custom hot end connected to the M16 threads of the Wellzoom that permits co-axial extrusion of thermoplastics and liquid metals in a similar format to that reported in previous work [22] but without the 3D printing capability. Only a single input and output for the liquid metal and polymer was used. A syringe pump was used to control the flow rates of the liquid metal when directly extruded, but we have also provided the capability of the needle to be open to the atmosphere or pressurized air in this work. The first design built at the University of Regina used a 22-gauge needle and a 1 mm diameter nozzle orifice for the wires shown in Table 1. By tuning the flow rate of the liquid metal, the temperature of the extruder, and the speed of the extruder, different relative internal diameters of stable liquid metal flow could be achieved. The control over the polymer flow rate is complicated by the resistance of the flow channels, viscosity of the polymer, and back pressure capability of the extruder itself, but the absolute maximum flow rate from the Wellzoom B is approximately 16 mm^3^/second based on the equipment datasheet.

Later versions of the extruder developed at the University of Alberta were set up to work with a more viscous polymer (G1645) which has a lower melt flow index than G1657 but also a lower durometer at room temperatures and generally higher performance in blown film applications, and it was hoped that drawing performance might be improved as well. For these materials, the centring insert was placed within the bottom M6 nozzle that had an internal diameter of 3 mm. The tip of the nozzle was modified to form a larger orifice (~1.8 mm diameter) or shark skinning defects that would become prominent from the shear forces during extrusion at lower temperatures. This extrudate size was also compatible with 3D printer extruders to feed into the thermal draw tower if needed. The needle was switched to 24-gauge and extrusion could be performed with air at different pressures. The fabricated tubes reported in this work were completed with the needle open to the atmosphere, and that provided highly consistent internal diameters. The system was used to prepare hollow SEBS tubes of both G1645 and G1657 to compare the injection of metal and drawing as separate steps. A schematic of the basic extrusion systems used is shown in Figure 1.

For subsequent reduction in the wire diameters, two operations were available—direct pulling from the extruder to reduce the wire diameter in a consistent process (see Appendix A), or a system for working with alternative form factors or previously extruded wires, called a thermal draw tower. The tower consists of several independent pieces of equipment designed to work together to preheat and draw down to specific sizes filaments, wires, and preforms. It is modular in design, to be functional with different input materials and form factors, but the core setup is shown in Figure 2. Feeding mechanisms were custom-built based on recovered 3D printer components, while the furnace used in this work was an ATS Series 3210 Split Tube Furnace and the filament was collected with a Filabot Filawinder. To thermally draw down materials, a draw-down ratio needs to be found that places a limit on how much a material can be reduced in a cross-sectional area in a single step. The stability of the polymer or liquid metal may impose limits on this operation so systemic studies will be left to future work after determining preferred general strategies in the production of stretchable wires. The initial investigation of strategies for drawing, separating, and integrating stretchable liquid metal core wires is the primary focus of this paper.

After fabrication, the wires were tested to determine how they would behave electrically. To examine the performance and durability of these designs, a test setup consisting of a linear positioner (RATTMOTOR part # CBX1605-100A) connected to an Arduino MEGA was used to record output resistances at different strain values. Multiple wires were tested to determine the variation in resistance, which could come from the internal variation in the cross-section or the contact resistances from the interconnection process. Calibration of wires is expected to be needed for long-term applications both due to the non-linear nature of resistance vs. displacement but also the inevitable variations in manufacturing and connection. However, this is not an insurmountable problem with appropriate training processes and detailed investigations [29].

### 2.2. Fabrication Process

Some wires were directly extruded and tested, while others were based on a metal-filled preform that was then heated and drawn down to different sizes. A complicating factor for the liquid metal wires is that for a sensor, using less liquid metal with a thin cross-sectional area is highly desirable, but if it is extremely thin in a fibre form factor, traditional techniques for connecting (like soldering, crimp connectors, or insertion of solid wires) are impossible due to either mechanical mismatch, thermal tolerances, or the precision of insertion being too demanding.

The production of thin liquid-metal-core thermoplastic elastomer wires can be performed with a variety of mechanisms, some as simple as pulling the extrudate melt immediately after extrusion, some by heating up a pre-fabricated thermoplastic wire and pulling it after it is close to the melting point, or technologies like thermal draw towers to provide a more controllable environment. Of these systems, the immediate pulling after extrusion is the conceptually simplest design, but normally results in only a single cross-section and is limited by the rapid cooling of the material as it is reduced in diameter which will ultimately fracture/fail if pulled too fast from the extruder nozzle. A particular challenge with some of the thermoplastic elastomers tested is that at a temperature close to but not at the melting point, they can become mechanically weak yet not very ductile. This aspect of polymer behaviour can pose greater challenges than drawing materials like polyethylene or polypropylene, which generally become more ductile as the temperature increases [31]. The thermal draw tower system is in theory the most repeatable and controllable but is designed primarily for larger initial diameters and complex internal features to be drawn down to micron or sub-micron sizes. The uniformity of drawing towers is set to provide a large amount of material manufactured with a consistent cross-section. As interconnections that have a larger end than the middle of the wire are mostly desired, this system is not appropriate for these particular wires reported but can be modified to work on pre-heating discrete sections of wires rather than large spools of continuous materials.

When co-axially extruding liquid metal and polymer simultaneously, it is important to control the flow rates of each to obtain a desired ratio of metal to polymer, and this is a function of feed rate, temperature, and nozzle dimensions. The simplest mechanism to control the flow resistance and flow rate for the equipment is to tune the temperature of the extruder nozzle itself, with lower temperatures leading to higher fluid resistance and lower flow rates of polymer. If the viscosity of the polymer is too low or the liquid metal core is too large in comparison to the polymer shell, unstable flow and droplets would occur, rather than stable jetting [22]. Initial wires were manufactured at the University of Regina and shipped for integration and testing at the University of Alberta. A variety of liquid metal core diameters were produced, but only the largest of these are shown in Table 1. Smaller cores and shells were possible to manufacture, but extraordinarily difficult to integrate with other wires. All of these samples show dimensions that are smaller than the actual nozzle/needle dimensions because they are being pulled at a rate of 70 mm/s after exiting the nozzle as per the setup in Appendix A. To test alternative manufacturing techniques, a modified extruder system at the University of Alberta produced hollow core fibres from G1645 at a variety of nozzle temperatures (Table 2), and the needle opened to the atmosphere to produce a more centred core of the tube and determine an acceptable temperature range for extrusion (Appendix A). The results were very consistent, between 180 and 205 °C, but roughening of the tube’s external diameter was observed at temperatures below 180 °C.

Gallium is a highly reactive metal, easily corroding and dissolving metals like aluminum or gold, and forming intermetallic compounds with materials like copper. Longer-term durability of interconnection to liquid gallium alloys has not been examined in the context of complex electronics but has been evaluated in other work where elevated temperatures were used to accelerate possible interactions with pure gallium and several candidate electrodes. While graphene and a few other materials have been reported as successful barrier layers to reduce gallium corrosion of electrodes, this step introduces far more complex processes and expected costs to what would ideally be a very low-cost manufacturing technique. Our preferred solution is to use electrode materials that are inherently non-corroding, but these are very limited based on a survey of academic literature. While many papers have successfully used copper/EGaIn interfaces for electrical connection [10,13,14], there are numerous reports of intermetallic compounds produced in this connection method [32] and it is not completely certain how these fare over long periods of time. To minimize potential issues, tungsten wires have been selected with a diameter of 0.025” (~0.635 mm), compatible with Dupont female connectors. Tungsten has two major advantages—it is one of the only reported metals that does not form an intermetallic compound with gallium, even at high temperatures [33], and it is highly electrically conductive, with approximately one-third the conductivity of copper. This outperforms some possible alternatives like stainless steel (which has electrical conductivity approximately 1/10th that of tungsten) which has been shown to partially resist gallium corrosion at high temperatures [1,34]. If only moderate temperatures are expected, however, and the resistance of the EGaIn wire is substantially higher than the stainless-steel interconnection, then stainless steel may be a more cost-effective alternative.

A desired wire connector diameter is selected based on the convenience of assembly in later steps and the maximum desired conductivity of the EGaIn wire per meter. For example, if a direct integration with a Dupont connector is desired, an inner diameter slightly smaller than 0.025” (~0.635 mm) should be the target—this is roughly the size of the male connectors for the Dupont system and will press-fit into the female connection. If the diameter is much larger, there could be higher leakage upon insertion of the wire, and if it is much smaller, the friction during insertion will make the process far more difficult. In practice, we have found that liquid metal cores much smaller than 30 gauge (~312 µm diameter) are too difficult for human operators to align and insert easily; therefore, we report only the systems that are inserting the Dupont-sized connectors even though other wires may work with this method. The 24-gauge needles used in the later extrusion system were found to produce core diameters slightly smaller than 0.56 mm (approximately the OD of that gauge size) when G1645 is extruded under tension from gravity with a 25 cm drop to the collection surface—permitting the diameter to naturally produce a semi-snug fit with the tungsten wires after insertion. While the natural friction between the SEBS and tungsten insert can provide a fairly strong mechanical interface, it is preferable to guard against small liquid metal leakage as well as improve the mechanical resilience at the interface. Tungsten wires may be dipped into a 1 M Na_2_CO_3_ solution or other acid or base just prior to insertion to help remove gallium oxides and ensure a better electrical connection between the tungsten and liquid metal [14]. After insertion and confirmation of electrical conductivity, it is feasible to connect the rest of the wire with heat shrink tubing or even a small amount of hot glue, which can bond relatively well to both SEBS and tungsten. If using heat shrink tubing, it must be only around the solid metal insert, or the heat and deformation of the tubing will permanently change the liquid metal core shape (see Appendix A).

If the liquid metal is injected into a previously extruded thermoplastic elastomer tube prior to thermal drawing, there are negligible deformations if only small pressures are applied. The circular cross-section and smooth internal walls generally prevent issues with trapped gas or uneven flow at the outset. Only for very low temperatures using G1645 was internal structuring of the tubes via sharkskin defects noticed and so filling these channels prior to thermal drawing was a straightforward process with predictable results (Figure 3).

Thermal drawing of very short segments is currently impractical with a thermal draw tower but can be approximated by a heat gun used for shrink tubing. A system available in the lab was found to melt the SEBS effectively and the use of a thermal couple in the stream of hot air indicated that the maximum temperature would be 275 °C. While not necessarily optimal, careful observation of the SEBS deformation and sag within the hot air stream can be used to determine when it is ready for drawing.

The direct drawing of liquid metal wires after extrusion has been described in other works, so only a description of current methods for taking smaller sections of previously extruded tubes and creating functional thin stretchable wires is provided here. A standard procedure for getting relatively consistent drawn-down tubes (without metal) is as follows (Figure 4):Starting materials: Co-axially extruded hollow SEBS (Kraton G1645 or G1657).Cut a 45 mm length of tube and mark the middle 15 mm length with a sharpie. Ensure the cuts are clean and flat.With a Mastercraft heat gun on level 2 setting, heat the 15 mm region for 10–15 s by holding the wire close to the side edge of the heat gun.Remove from heat and promptly stretch/draw the wire until the marks are 150 mm apart (a ruler nearby provides a guide). The draw should be gentle and consistent. Faster draw speeds will create thinner diameter wires, but risk breaking during drawing. As a rule of thumb, drawing the wire from 15 mm to 150 mm should take <5 s.

**Figure 4 micromachines-14-02227-f004:**
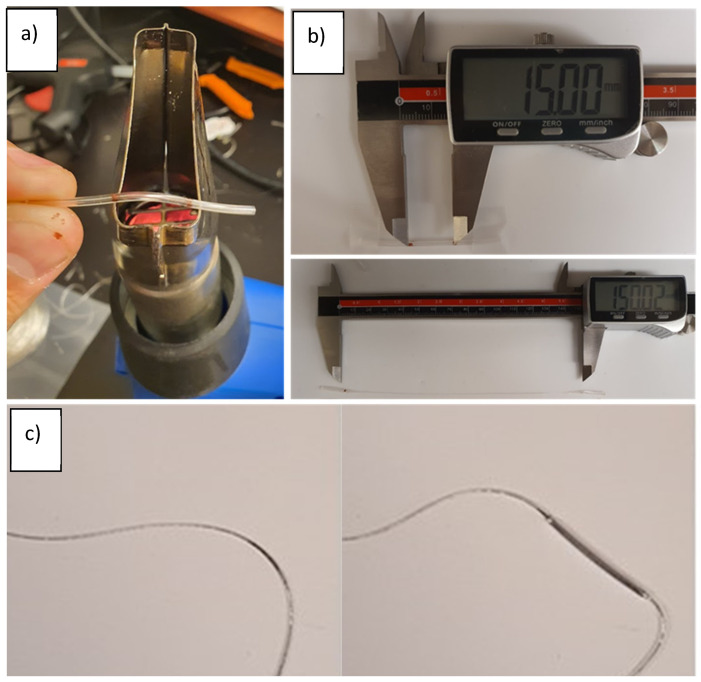
(**a**,**b**) Hollow G1645 SEBS tubes drawn out to 10× their original length plus large end segments. EGaIn can be injected into these designs after drawing—subject to pressure limits for the tubes themselves. (**c**) Images from an injection failure that inflates the SEBS in a snap-through defect with excessive pressure input. More common failures are incomplete metal filling or the tube being ejected from of the injection port. These failures impose an upper limit on the pressures that can be used to fill channels and ultimately the size of pre-drawn channels that can be attempted.

Completing the same process can work with liquid metal already filled in the core for these wires but we have found that for short segments at least, one end of the wire should be open to the atmosphere, or the metal may expand and burst the tube prior to drawing. Additionally, overheating the liquid-metal-filled wire and drawing it down too quickly can result in metal droplet breakup and discontinuous electrical paths (See Appendix A).

With improved techniques, thermal drawdown can make very long thin tubes that are still capable of working with an injection of metal after the drawing process. Tubes up to 50 cm long with internal diameters of approximately 50 µm have been successfully filled with liquid metal but lengths beyond that are more feasible with the thermal drawing of pre-filled tubes. While the diameters may be still further reduced in size, handling of the final fibres becomes increasingly challenging and a different inner-to-outer diameter ratio could be chosen to start, or protective coatings or strain-limiting layers could be introduced in future variations. Figure 5 shows the different stages of the production and draw-down process for liquid metal wires compatible with Dupont connectors.

## 3. Results and Discussion

### 3.1. Cutting and Sealing Strategies for Liquid Metal Continuous Wires

Whether a liquid-metal-core wire is produced in one step or through hollow tube production followed by injection, strategies are needed for cutting the wires to length and either exposing the liquid metal or containing it within the wires for leak-free storage. Cleaning strategies for minimizing or eliminating contamination of wire exteriors and cutting tools are also needed for the effective production of wires of the desired shape and size. In this work, we started by evaluating different cutting tools available in our lab for their precision, utility, and ability to cut without excessive metal loss or cross-contamination of the cutting tools. The qualitative results are shown in Table 3 and a rotary cutter provided the best general results. For the purposes of sealing the ends of wires for storage and potentially discovering improved methods for bonding the materials to other components of soft robotics (as an example), several options were tested, as listed in Table 4. Attractively, the use of ultrasonic welding appears to be very viable for large-scale operations as the applied pressure and application of energy can be precisely controlled. Our available options include either a large-sized welder (Branson Ultrasonics 2000X system, Branson Canada, Toronto, ON, Canada) or a hand-held ultrasonic knife from MicroMark (Wondercutter S, Berkeley Heights, NJ, USA). To ensure welding and separating of the SEBS rather than clean cutting, a filed down and rounded knife edge was used for the Wondercutter. Our best option for cutting wires for integration with solid metal connectors was the rotary blade and the best solution for welding/sealing the longer liquid metal wires into shorter segments was the ultrasonic knife. The typical results of those tools are shown in Figure 6 while the less optimal options are shown in Appendix A.

### 3.2. Filling and Drawing Strategies for Liquid Metal Wire Segments

The production of a variable cross-section liquid metal wire is challenging to complete with a high degree of repeatability. While the concept is quite simple and has even been shown to work with plastically deformable materials at room temperature [35], there are major challenges involved in the consistency of the formation, including undesirable necking, breakup of metal into droplets if the viscosity of the polymer is too low, and fracture of the polymer if the tension is too high. Regardless, these methods when optimized are excellent for creating complex multi-material filaments and reducing their cross-section to extremely small sizes (potentially in the nm scale) [23,31] in a consistent fashion if a uniform cross-section area is desired. Smaller liquid metal cores are desirable for two major reasons—increasing resistance for sensor applications to minimize the influence of contact resistance in other parts of the circuit, and making their production economically more viable.

As a conservative estimate, a liquid metal wire with a 0.025” (~625 µm) EGaIn core diameter could consume approximately 0.31 mL of EGaIn per meter of wire produced, which corresponds to just less than 2 g of material. At a cost of 300–600 USD/kg depending on source [21], this is approximately 0.6–1.2 USD/m. The electrical resistance of a wire would be equal to the length/cross-sectional area of a wire multiplied by the resistivity of the material. Assuming a resistivity of EGaIn of 29.4 × 10^−6^ Ω cm, the 1 m long wires with 625 µm diameter have a resistance of approximately 1 Ω/m, which is acceptable for powering systems but too small to work effectively as a sensor in most cases (resistance may be dominated by other components or contact resistance in the circuit). If thermally drawn down to reduce the inner diameter to 25 µm from the original core size (a 25× reduction), the resistance per meter will be 625 Ω, and the process will create a fibre 625 m long, reducing costs to a fraction of a cent per meter. Preferably, the wire design aims for a larger physical size at either end to facilitate integration with Dupont connectors (or similar reversible connections). Unfortunately, conventional methods for injecting liquid metal at these new channel dimensions encounter challenges. A wire drawn out to four times its original length (and assuming constant volume) would have its radius reduced by half, significantly affecting the flow resistance near the end of the injection process.
$$R_{flow}=\frac{∆P}{Q}=\frac{8\mu L}{\pi r^{4}}$$

For this formula, Δ*P* is the pressure difference, *Q* is the volumetric flow rate, *µ* is the viscosity, *L* is the channel length, and *r* is the channel radius. Assuming laminar flow, equal volumetric flow, and considering flow resistance just before the liquid metal completes the injection process, the resistance to flow increases by the third power of the draw ratio. As a result, a wire drawn to twice its original length would have 8× higher required pressure to fill for the same volumetric flow rate.
$$V=\pi r_{0}^{2}L=\pi r_{1}^{2}*(2L)$$
$$r_{1}=\sqrt{\frac{1}{2}}r_{0}$$
$$R_{flow}\alpha \frac{L}{r^{4}}R_{1}=\frac{2}{{\sqrt{0.5}}^{4}}R_{0}=\frac{4}{0.25}=8R_{1}$$

This analysis solely considers fluid viscosities while disregarding other complicating factors, such as the oxide layer and Laplace pressure, which would also introduce an additional source of resistance to filling. The ability to inject metal into extremely narrow and long channels will be limited by the amount of pressure that can be applied before the rubber tubes themselves may expand (Figure 4c) or lose contact with the injection nozzle.

During the testing of the wires that are produced with this basic fabrication technology several surprising results were obtained. The first was the relative difficulty of thermal drawing with G1645 compared to G1657. The lower melt flow index of G1645 was originally hypothesized to be more useful in maintaining stability during thermal drawing, but it would frequently show relatively weak strength unless heated well above its melting point. The draw furnace, or hot air guns used to prototype with this polymer were set at 275 °C to achieve good results, which was a surprise compared to what was initially expected. The possible draw-down ratio of a polymer is not solely determined by temperature and molecular weight but by other factors as well including strain rate and exact polymer microstructure [31]. The more liquid-like G1657 melt applied lower forces on the non-molten polymer during drawing at equal temperatures to the G1645, so it can be easier to work with during drawing operations. Another surprise was that for very small tubes, the electrical conductivities of G1645 were more easily permanently changed than for G1657. G1657 has a higher durometer and is noticeably less self-adhesive than G1645. Even for larger diameter fibres, it was observed that collapsing the tubes via a pinch could create a semi-permanent bond that could block the flow of material, and if this was performed with liquid metal as the core, it could interrupt the conductor and lead to an open circuit (Figure 6a). This vulnerability of the G1645 fibres to complete collapse is another reason that their use may ultimately be less desirable for stretchable electronics than first assumed. Despite these challenges, the mechanisms for creating highly stretchable and almost imperceptible strain sensors with these drawn-down wires hold great promise for the eventual integration of electrical versions of optical sensors or pneumatic controls within soft robotic systems, haptics, or wearable electronics [16], as well as damage detection and other functionalities. A simple test of repeatability was set up and shown in Figure 7.

The choice to connect the liquid metal wires to tungsten ends of 0.025” in diameter was proven to work repeatedly when connected to common electronic components like breadboards, Arduino boards, and simple Dupont extender wires. The same liquid metal wires can be easily connected and disconnected to prototypes including Pneunet actuators, linerar stages, and R3VAMPs actuators. The extremely thin design of the wires requires very little force to hold them in place on the external surfaces of materials, and double-sided tape, hot glue, or epoxies are all options to bond the wires temporarily or permanently. Importantly, Dupont connectors are not designed to be load-bearing so if large axial loads are applied to liquid metal wires they will pull out of these connectors. We have successfully crimped other end connectors like JST-SM to the tungsten ends, however, and other reinforcements can be possible in the future, but in most cases, the material limits of the SEBS would provide the ultimate strength of the part rather than the connector itself.

## 4. Improvements and Applications

The philosophy behind developing this work is to provide inexpensive and scalable manufacturing processes for the future integration of sensorized soft robotic systems, wearable electronics, and intelligent materials. Infrastructures for large-scale production are being set up but in the interim, the basic guidelines on what is possible to achieve with thermoplastic extrusion and drawing for ultra-fine stretchable electronically functional fibres were investigated. Thermoplastic elastomers have enormous benefits in production capabilities in comparison to silicone rubbers and the manufacturing, cutting, and assembly technology can be applied to a much broader range of materials in the future. Exceptionally fine wires may be woven, cast into, or bonded to a variety of materials to sensorize them and be directly compatible with standardized electrical connectors so that non-experts may be able to access these materials and allow for more accessible technology use. As a proof of concept, we have sensorized some of our previously reported soft robotic actuators—the R3VAMP [36] and a 3D-printed pneunet actuator made of G1657 [37]. R3VAMP actuators are based on 3D-printed infill settings that can then actuate under vacuum in linear motion, bending, or jamming while the pneunet is more traditionally designed but printed with the same type of thermoplastic elastomers used in this work. A pre-tensioned liquid metal wire can be fed through the R3VAMP or directly bonded onto the actuators with adhesives as shown in Figure 8. For this demonstration, a post-manufacturing insertion or attaching of a liquid metal strain sensor that is pre-stretched in the system adds negligible stiffness and force to the original actuator and permits the measurement of displacement electrically during vacuum-driven operation. Extending this mechanism to more wires located in different areas combined with training of the system can permit complete proprioception for the robotic design [29]. Attractively these SEBS tubes are chemically compatible with materials like polyethylene, polypropylene, and polystyrene among others, so thermal welding and connecting them to other thermoplastic soft robotic systems or wearables is very feasible in the future. A description of the R3VAMP fabrication method is found in the Appendix A.

Other improvements that we expect to pursue in the future would be an improved insertion process for tungsten wires. A stepped or tapered inserted portion should increase the contact area between the liquid metal and the solid wire, and make it less likely to trap any gas within the rubber tubes during the insertion process. An ultimate goal would be a system closer resembling a crimp connector, which could just be pressed onto the rubber wire and contact the liquid metal inside through puncturing or other automatic mechanisms. For the time being, those efforts have been unsuccessful but they would be a major goal to enhance viability for large-scale manufacturing.

## 5. Conclusions

To enable liquid-metal-enabled electronics to make a substantial impact outside of academia it is not enough to focus solely on device development, and one needs to consider the whole system. Just as modern electronics cannot work without consistent and reliable connection systems, liquid-metal-enabled electronics must include ways to standardize connections to ensure that consistent results and higher-level integration with complex systems can be achieved. By focusing on a protocol to integrate liquid metal wires to existing reusable connections for traditional electronics prototyping, this work advances a vital niche within liquid metal and soft robotics research. Choosing to initially focus on Dupont-type connectors immediately allows liquid metal wires for sensors, actuators, and electrical connections to work with Arduino, Raspberry Pi, breadboards, and other rapid prototyping systems for electronics, greatly simplifying iterations and prototypes, enhancing the future complexity of designs through reliable building blocks and integration. Our current system allows the rapid production of liquid metal strain sensors that can plug directly into Dupont connectors for quick integration and modification of sensorized elements, and shows a post-manufacturing integration with soft robotic actuators as a first demonstration, but may in future also be integrated with electronic sensing skins, neuroprosthetics, and stretchable antennas. Future improvements will determine automation options for completing the non-uniform drawing process for variable cross-section liquid metal wires and 3D printing more complex preforms for multi-functional liquid metal electronic systems.

## Figures and Tables

**Figure 1 micromachines-14-02227-f001:**
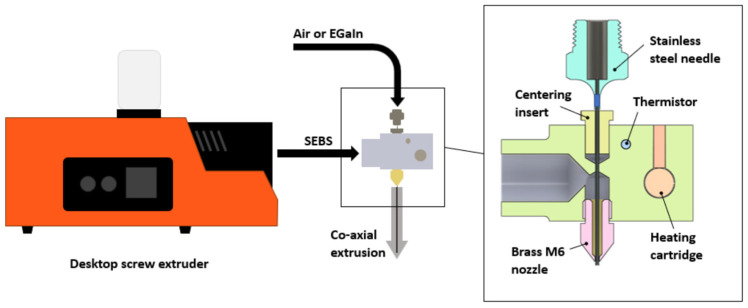
General co-axial extrusion system for hollow or liquid-metal-filled SEBS tubes.

**Figure 2 micromachines-14-02227-f002:**
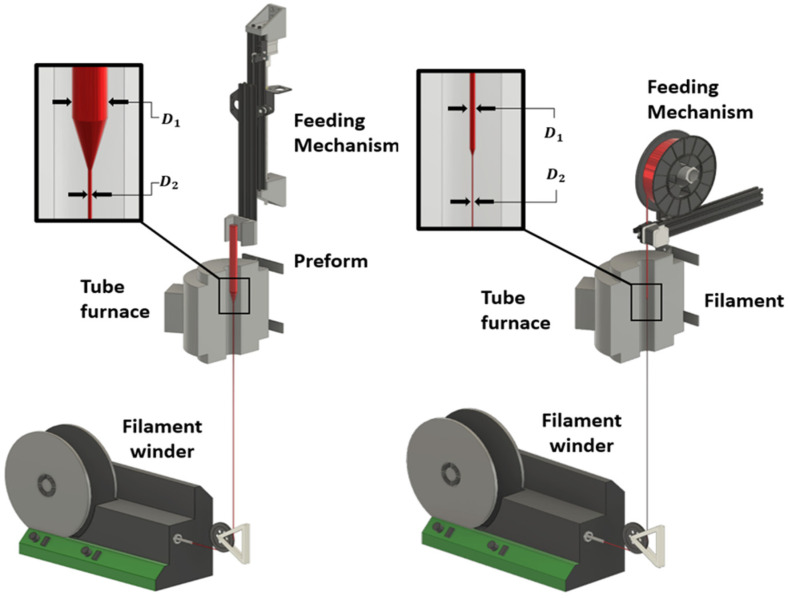
Basic thermal draw tower for 3D-printed preforms (using a z-axis positioner) or thermoplastic filaments (using a repurposed stepper motor extruder system) to be drawn down to smaller diameters. The maximum draw-down ratio (D_1_^2^:D_2_^2^) is a function of polymer properties, temperature, and heating profile. Very small fibres may be manufactured through multiple draw stages.

**Figure 3 micromachines-14-02227-f003:**
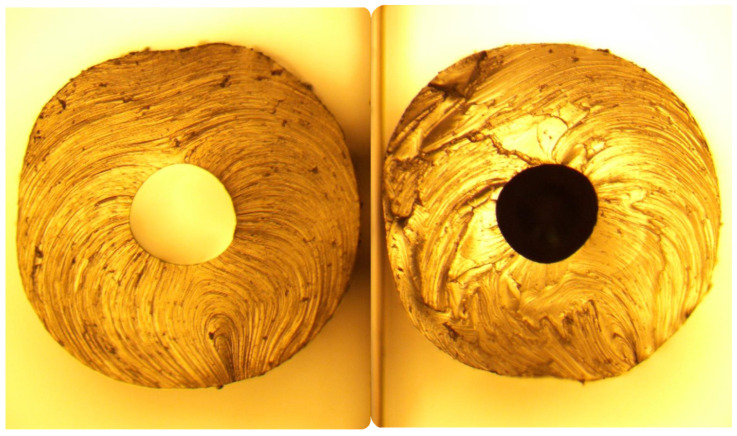
Composite image of the cross-section of a G1645 tube (extruded at 175 °C) prior to and after filling with liquid metal. No substantial distortion is seen so metal dimensions can be assumed to be equal to the original channel dimensions.

**Figure 5 micromachines-14-02227-f005:**
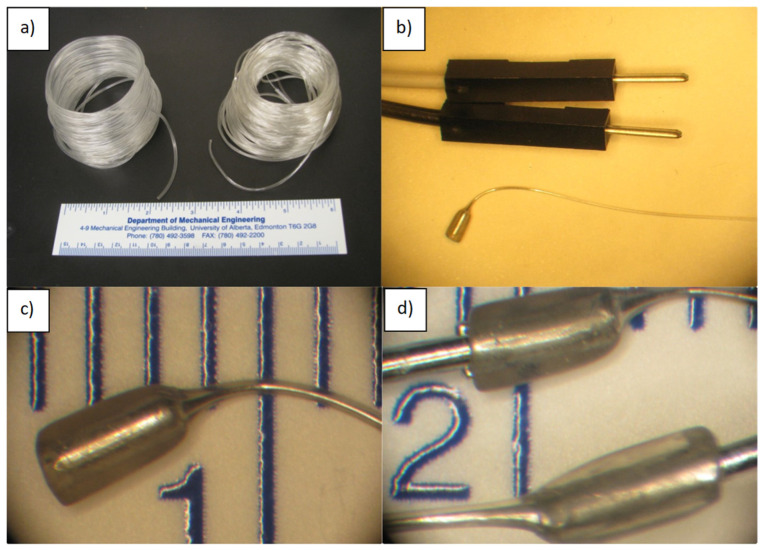
(**a**) SEBS tubes (G1645 and G1567); (**b**) thermally drawn and metal-filled G1657 SEBS tube next to commercial Dupont connectors to show scale; (**c**) close-up view of the interface of a thermally drawn wire that is reduced to approximately 1/6th its original diameter on top of the mm scale of a ruler; (**d**) inputs and outputs connected with solid metal connectors.

**Figure 6 micromachines-14-02227-f006:**
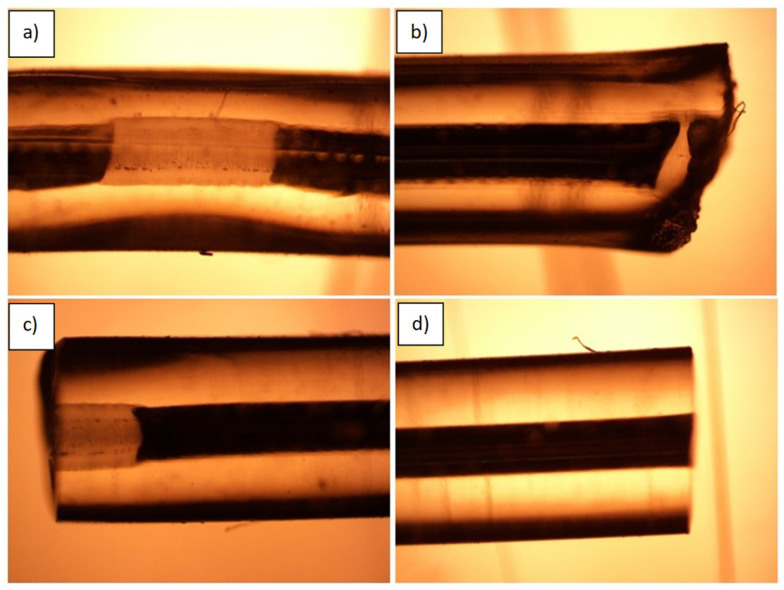
Microscope images of G1645 tubes filled with EGaIn that have been cut or bonded with different techniques. (**a**) Tube pinched with pliers to create electrical discontinuity within the wire—SEBS is bonded to itself, but metal oxide residue shows the outline of the previously filled channel. (**b**) Tube that has been cut and welded shut by a blunt ultrasonic knife. EGaIn is permanently sealed in the channel and no leakage occurs. (**c**) Previously pinched channel in (**a**) has been cut with a rotary blade. The EGaIn remains pinched off within the channel. (**d**) EGaIn channel is directly cut with a rotary blade. The EGaIn is stabilized by the oxide but open to air and the cut is very clean.

**Figure 7 micromachines-14-02227-f007:**
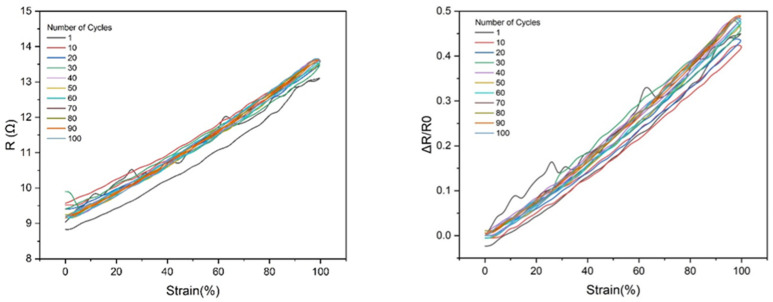
Absolute resistance and change in resistance for a liquid metal wire tested for 100 cycles. The strained portion was smaller than the full wire sample and the ends were wrapped around circular pillars to avoid unwanted pinch-off during the tests. Strain is measured only as a portion of the wire that is stretched. The very first cycle shows much more resistance variation than the others. This could be due to oxides breaking and reforming in the wire, or the influence of the Mullins effect from the large strain of the rubber.

**Figure 8 micromachines-14-02227-f008:**
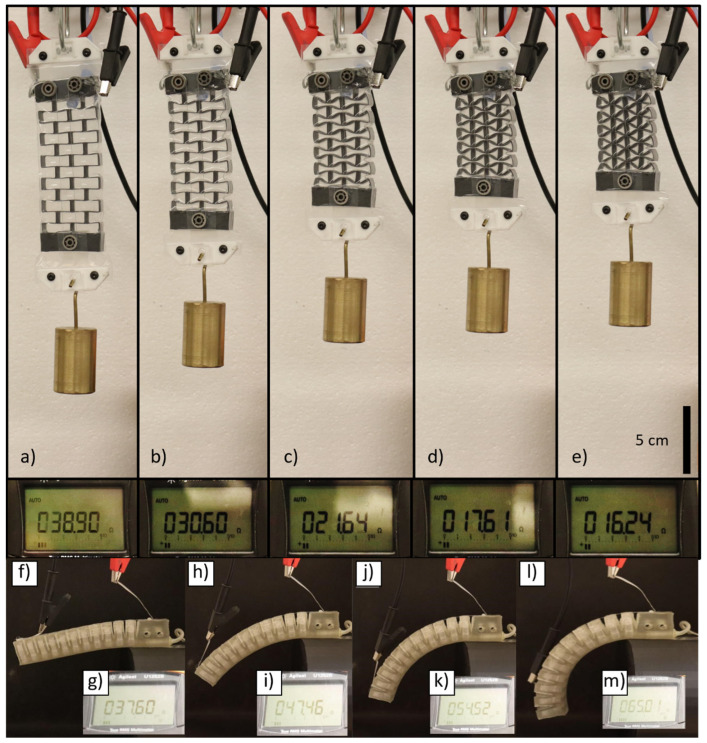
Two classes of thermoplastic soft robotic actuators previously reported: R3VAMPs [36] and Pneunets [37] sensorized with liquid metal wires after their original fabrication. (**a**–**e**) R3VAMPs actuation from 0 to −40 kPa in 10 kPa increments, with the bottom reading as the resistance in Ω of the liquid metal wire fed through the pulleys secured to the outside of the sleeve. (**f**,**h**,**j**,**l**) A thermoplastic pneunet with a liquid metal wire bonded to the top portion which extends during bending motion while resistance corresponding to each is shown in (**g**,**i**,**k**,**m**), respectively.

**Table 1 micromachines-14-02227-t001:** G1657 wires with LM core produced with various flow rates and temperatures. Cross-section images are used to find the core and shell diameters produced using a 22-gauge needle in a 1 mm diameter nozzle. Identically set flow rates for LM can result in different actual flow rates if the resistance to flow is high due to low temperatures or small nozzle sizes. These wires were pulled from the extruder at a speed of 70 mm/s to reduce their diameters.

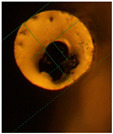	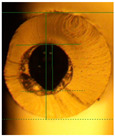	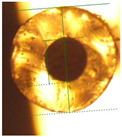	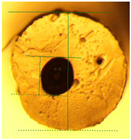
ID: 153 µm	ID: 307 µm	ID: 289 µm	ID: 256 µm
OD: 518 µm	OD: 707 µm	OD: 699 µm	OD: 780 µm
LM: 15 × 10^5^ µL/s SEBS: 4.50 mm^3^/s Temp: 190 °C	LM: 25 × 10^5^ µL/s SEBS: 5.50 mm^3^/s Temp: 210 °C	LM: 20 × 10^5^ µL/s SEBS: 5.50 mm^3^/s Temp: 210 °C	LM: 20 × 10^5^ µL/s SEBS: 6.50 mm^3^/s Temp: 190 °C

**Table 2 micromachines-14-02227-t002:** G1645 extruded hollow tubes without metal showing consistency of internal diameter and outer dimension from a 1.8 mm diameter nozzle and 24-gauge needle for a variety of extrusion temperatures. SEBS was collected after free-falling 25 cm from the extruder in these cases. Overall dimensions across multiple samples were 1760 ± 60 µm OD and 490 ± 40 µm ID.

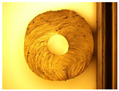	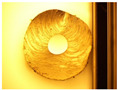	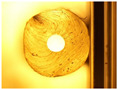	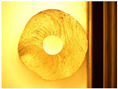
ID: 540 µm	ID: 510 µm	ID: 480 µm	ID: 455 µm
OD: 1840 µm	OD: 1770 µm	OD: 1720 µm	OD: 1730 µm
Temp: 175 °C	Temp:180 °C	Temp: 190 °C	Temp: 195 °C

**Table 3 micromachines-14-02227-t003:** Cutting Tool Comparison.

Tool	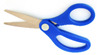 Scissors	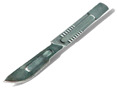 Surgical knife	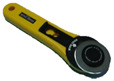 Rotary Blade
Pros	Easily accessible	Cheap and easily replaceable blades. Can cut on top of paper soaked with acid or base to remove metal oxides easily.	Extremely easy to roll over wires to cut cleanly; can cut on top of paper soaked with acid or base for removing metal oxides easily.
Cons	Most commercial versions can have difficulty cutting very stretchable rubber, shearing can release substantial quantities of liquid metal	Must be used on appropriate substrates to avoid damaging the blade, and must be used in a chopping motion to cut very flexible rubber without shearing the wire.	More expensive and less easy to replace blades than in a surgical knife.

**Table 4 micromachines-14-02227-t004:** Sealing tool comparison.

Tool	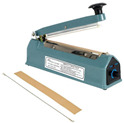 Impulse sealer	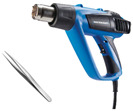 Hot gun and tweezers	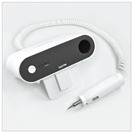 Ultrasonic knife
Pros	Easily accessible, fast	Can pinch off and separate in one step	Very easy to achieve consistent results with little metal loss. Can press first then apply ultrasonic energy to fuse thermoplastic ends together.
Cons	Must apply pressure first, then heat up for consistent results. The bonded area is relatively wide and unevenly structured due to the Teflon fabric on the impulse sealer. Extremely hard to clean if metal leaks.	Difficult to obtain repeatable results, chance of burns, highly skill-dependent with regard to timing	Relatively expensive, must modify knife to be blunt over the area of cut-off/welding to seal liquid metal rather than expose it, can overheat if used improperly.

## Data Availability

Supporting information is available from the authors upon reasonable request.

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
