# Peer review of "Towards High Efficiency and Rapid Production of Room-Temperature Liquid Metal Wires Compatible with Electronic Prototyping Connectors"

_micromachines, 2023, doi:10.3390/mi14122227_

Round 1
Reviewer 1 Report
Comments and Suggestions for Authors
In this work, the authors developed a system allowing the rapid production of liquid metal. Several key challenges have been addressed, including leak-free operation, minimal gallium corrosion of other electrode materials, low liquid metal consumption and high production rates. The work is done well and worth publishing in this journal. The referee suggest its publication after considering following issues.
1 The authors should state the advantages of this system compared to others.
2 For corrison, the authors should consider the acidity of the system and the electrochemical potentials should be provided.
further comments on the manuscript. I ask the authors to add measurement on flexibity of the materials, such as stress strain curves. More characterizations such as SEM, TEM, should be provided.Comments on the Quality of English Language
No
Author Response
In this work, the authors developed a system allowing the rapid production of liquid metal. Several key challenges have been addressed, including leak-free operation, minimal gallium corrosion of other electrode materials, low liquid metal consumption and high production rates. The work is done well and worth publishing in this journal. The referee suggest its publication after considering following issues.
1 The authors should state the advantages of this system compared to others.
Thank you for the question. We have highlighted the primary advantages in the introduction with new sentences as follows:
This work demonstrates for the first time strategies for the continuous production of liquid metal wires and their subsequent reduction in size to create microscale feature that are still compatible with standard electronics prototyping connectors. In particular, the ability to have larger diameter liquid metal core rubber wire of substantial length be cut and sealed directly without loss of internal liquid metal is extremely important for eventual mass production as hand injection or manual operations can be minimized or eliminated with future development.
2 For corrison, the authors should consider the acidity of the system and the electrochemical potentials should be provided.
This is an excellent question, although the acidity of the system is not capable of being directly measured with our experimental equipment and we are not currently running it in an electrolytic solution, the direct contact of different metals is a source of issue. The the electro potential of gallium (-.56 V in for Ga ⇌ Ga3+ + 3e-) and Indium (-0.3382 In ⇌ In3+ + 3e-)) and tungsten 0.1 V W ⇌ W3+ + 3e- are found from the CRC handbook.
Lide, David R., ed. CRC handbook of chemistry and physics. Vol. 85. CRC press, 2004.
If the reaction is not creating the 3+ values for In and Ga the electropotential difference could be lower that needed to create spontaneous alloying, but we can’t be absolutely sure of the mechanism.
further comments on the manuscript. I ask the authors to add measurement on flexibity of the materials, such as stress strain curves. More characterizations such as SEM, TEM, should be provided.
More characterization tests are planned. For the timeframe provided we didn’t have time to do new experiments, but examples of some SEM images of previous liquid metal wires and characterization of SEBS are found in the following references:
Khondoker, Mohammad AH, Adam Ostashek, and Dan Sameoto. "Direct 3D Printing of Stretchable Circuits via Liquid Metal Co‐Extrusion Within Thermoplastic Filaments." Advanced Engineering Materials 21.7 (2019): 1900060.
Bschaden, Benjamin Simon. "Developing Design Guidelines for Improved Gecko Inspired Dry Adhesive Performance." (2014). https://era.library.ualberta.ca/items/ba71f8a6-cf16-4396-b7bc-419fa83ff8eb
Reviewer 2 Report
Comments and Suggestions for Authors
I would like to invite your response to the following comments:
1-Please provide the the corresponding complete expressions for JST in abstract, PDMS, MEMS...etc
2- Please modify the "liquid metal" to "room-temperature liquid metal"
3-Please provide the detailed technical information of fabrication of your alloy, including the furnace atmosphere, the protection of alloy production etc.
Comments on the Quality of English LanguageThe language of manuscript is acceptable
Author Response
1-Please provide the the corresponding complete expressions for JST in abstract, PDMS, MEMS...etc
We have defined polydimethylsiloxane (PDM), Japan solderless terminal (JST), microelectromechanical systems (MEMS)
2- Please modify the "liquid metal" to "room-temperature liquid metal"
We have changed the terminology in the title and the first time it is used in the abstract. The number of times the term was used in the paper made the document very wordy so we hope that this would suffice for clarity in the context of the work.
3-Please provide the detailed technical information of fabrication of your alloy, including the furnace atmosphere, the protection of alloy production etc.
The alloy can actually be made at room temperature and no furnace is needed, but we described the original process in the first part of the materials and methods section as follows:
Gallium and Indium were obtained from rotometals.com and combined at 75.5% wt% Gallium to 24.5% wt Indium by melting the Gallium in a container surrounded by boiling water and adding Indium while stirring until no solid metal remained.
We have added the following sentence.
The oxide was avoided by extracting material from the sub-surface of the alloy with a syringe prior to use.